

# Identification of a prognostic signature of nine metabolism-related genes for hepatocellular carcinoma

Chaozhi Tang[1,*], Jiakang Ma[2,*], Xiuli Liu[3] and Zhengchun Liu[4]

[1] Department of Urology, The First Affiliated Hospital of China Medical University, Shenyang, Liaoning, China
[2] Department of Oncology, The Second Affiliated Hospital of Zhengzhou University, Zhengzhou, Henan, China
[3] Department of Oncology, Affiliated Hospital of Guilin Medical University, Guilin, Guangxi, China
[4] Department of Radiation Oncology, Affiliated Hospital of Guilin Medical University, Guilin, Guangxi, China
[*] These authors contributed equally to this work.

## ABSTRACT

**Background**. Hepatocellular carcinoma (HCC) is the fifth most common cancer. Since changes in liver metabolism contribute to liver disease development, it is necessary to build a metabolism-related prognostic model for HCC.

**Methods**. We constructed a metabolism-related-gene (MRG) signature comprising nine genes, which segregated HCC patients into high- and low-risk groups.

**Results**. The survival rate (overall survival: OS; relapse-free survival; and progression-free survival) of patients in the low-risk group of The Cancer Genome Atlas (TCGA) cohort was significantly higher than that of patients in the high-risk group. The OS prognostic signature was validated in the International Cancer Genome Consortium independent cohort. The corresponding receiver operating characteristic curves of the model indicated that the signature had good diagnostic efficiency, in terms of improving OS over 1, 3, and 5 years. Hierarchical analysis demonstrated that the MRG signature was significantly associated with better prognosis in male patients, patients aged $\leq 65$ years, and patients carrying the wild-type *TP53* or *CTNNB1* genes. A nomogram was established, and good performance and clinical practicability were confirmed. Additionally, using the GSE109211 dataset from the Gene Expression Omnibus database, we were able to verify that the nine genes in this MRG signature had different responses to sorafenib, suggesting that some of these MRGs may act as therapeutic targets for HCC.

**Conclusions**. We believe that these findings will add value in terms of the diagnosis, treatment, and prognosis of HCC.

## INTRODUCTION

Hepatocellular carcinoma (HCC) is the fifth most common cancer and the most common type of hepatobiliary cancer, with more than 500,000 new cases diagnosed each year and an

Corresponding author
Zhengchun Liu,
18624879717@163.com

annual mortality rate of 250,000 (*Schlachterman et al., 2015*; *Singal & El-Serag, 2015*; *Steel et al., 2007*). In North America and several European regions, the incidence and mortality rates of HCC have been rising, and in the United States alone, the incidence of HCC has more than doubled in the past 20 years (*Singal & El-Serag, 2015*). Currently, only 46% of HCC cases are diagnosed early, with most cases being diagnosed too late to allow for successful treatment (*Njei et al., 2015*). The 5-year overall survival (OS) of all stages of HCC is 12%, increasingly making it the most important cause of cancer-related death (*Hilmi et al., 2019*). In modern clinical practice, the major risk factors for HCC are chronic hepatitis C virus (HCV) or hepatitis B virus (HBV) infection, heavy drinking, diabetes, and nonalcoholic fatty liver (*Budny et al., 2017*; *Hsu et al., 2015*; *Levrero & Zucman-Rossi, 2016*). In particular, active HCV and HBV contribute the most to the burden of global HCC. Several advances related to HCC prevention and early detection and diagnosis have proven to be effective and have led to a reduction in the incidence of HCC and the mortality associated with it. Nonetheless, there are several common challenges in detecting and treating HCC, including limited awareness about high-risk patients, limited availability of effective and validated risk-stratification measures, and high costs of monitoring at-risk populations.

HCC is a highly angiogenic solid tumour characterised by cell cycle disorders, abnormal angiogenesis, and escape from apoptosis (*Bhagyaraj et al., 2019*; *Rebouissou & Nault, 2020*). The molecular pathogenesis of HCC is complex and involves a variety of genetic and epigenetic changes, chromosomal aberrations, genetic mutations, and altered molecular pathways (*Farazi & DePinho, 2006*). Metabolic alteration is one of the important features of tumours. The liver is an important hub for metabolism of the three major nutrients— sugar, lipid, and amino acids—in the body. HCC exhibits a variety of characteristic metabolic changes, such as increased oxygen glycolysis, de novo fat synthesis, glutamine consumption, and oxidative metabolic imbalance, which provide energy to the rapidly growing and proliferating tumour cells. The process of metabolic alterations in tumours is regulated by multiple factors, such as changes in metabolic enzyme activity, abnormal gene expression, and dysfunctional signal transduction pathways. Many clinical parameters currently used to assess liver function reflect changes in enzyme activity and metabolites. In fact, the difference in glucose and acetate utilisation has been used as an effective clinical tool to stratify HCC patients. In addition, elevated serum lactate levels can distinguish HCC patients from healthy individuals, and serum lactate dehydrogenase is used as a prognostic indicator for patients with HCC during treatment. It is thus necessary to build a metabolism-related prognostic model for HCC (*De Matteis et al., 2018*; *Nakagawa et al., 2018*; *Pope 3rd et al., 2019*). Changes in metabolic pathways, which are driven by oncogenes, are recognised as cancer markers, and such changes provide cancers with a selective advantage for tumour growth, proliferation, and metastasis (*Berndt et al., 2019*; *De Matteis et al., 2018*; *Huang et al., 2014*; *Kim et al., 2019*). *TP53* and *CTNNB1* are two genes that are most prone to mutations in HCC, and have received continuous research attention because of their involvement in events that dominate tumour development and progression (*Calderaro et al., 2017*; *Cancer Genome Atlas Research Network. Electronic address & Cancer Genome Atlas Research, 2017*; *Zucman-Rossi et al., 2015*). Treatment with

sorafenib, the first approved systemic therapeutic agent for HCC, has shown significant improvements in survival outcomes of HCC patients (*Ogasawara et al., 2018*; *Pinter & Peck-Radosavljevic, 2018*). It inhibits cell growth, induces apoptosis, and downregulates the anti-apoptotic protein Mcl-1 by targeting a variety of protein kinases (*Tai et al., 2013*).

In this study, we constructed a prognostic model based on a metabolism-related gene (MRG) signature comprising nine genes. Further, we examined the associations among this signature, sorafenib use, and common clinical characteristics such as sex, age, and the presence or absence of mutations in *TP53* and *CTNNB1*. The findings from this study will supplement the existing literature on HCC by providing useful data regarding molecular diagnosis and precision treatment and will help in screening drug targets for HCC.

## MATERIALS & METHODS

### Data processing and extraction of differentially expressed MRGs

For the discovery cohort ($n = 374$ patients), fragments per kilobase of exon per million mapped reads (FPKM) data and the corresponding clinical information related to gene expression were downloaded from The Cancer Genome Atlas (TCGA). For the validation cohort ($n = 243$ patients), the original data and corresponding clinical data related to gene expression were downloaded from the Liver Cancer-RIKEN, Japan (LIRI-JP) project of the International Cancer Genome Consortium (ICGC). For analysis of OS, relapse-free survival (RFS), and progression-free survival (PFS) in the discovery cohort, and that of OS in the validation cohort, we used data from 343, 301, 337, and 230 HCC patients, respectively; these patients had been followed up for 30 days or more, and patients who were likely to die of lethal complications (heart failure and haemorrhage) other than HCC were excluded from the study. The probes were annotated using *Homo sapiens* GTF files in the Ensembl database (http://asia.ensembl.org/index.html).

We selected all the pathways related to metabolism among 186 pathways from c2.cp.kegg.v7.0.symbols.gmt. In total, 944 MRGs were obtained, of which 918 MRGs were common between TCGA and ICGC. TCGA and ICGC MRG expression matrices were background-corrected using the R package "sva" with the "limma" package to identify differentially expressed MRGs. Genes having $|\log2FC| > 2$ and $P$-value $< 0.05$ were selected for further analysis. The genes whose expression values were more than 0.5 and thus subjected to log2 transformation are depicted in a heat map. Finally, 54 differential MRGs were selected. The GSE109211 dataset was downloaded from the GEO database (https://www.ncbi.nlm.nih.gov/geo/) and contained 140 cases of HCC (67 patients who received sorafenib and 73 patients who received placebo). This dataset was used for gene annotation based on the GPL13938 platform (Illumina HumanHT-12 WG-DASL V4.0 expression beadchip).

### Identification of a prognostic signature based on differential MRG expression

After screening for differentially expressed MRGs, we performed univariate Cox regression analysis to identify prognostic differentially expressed MRGs. A $p$-value $< 0.05$ was considered statistically significant. Next, the "glmnet" package was used to implement the

LASSO Cox regression model. All regression coefficients, including the coefficients of many unrelated features, were reduced to zero in LASSO, precisely according to the adjustment weight λ. The best λ was selected based on the minimum cross-validation error. Finally, we calculated the risk coefficients of the genes associated with the LASSO Cox regression model and estimated the median risk, using the "survminer" package in R.

## Survival analysis

The risk scores of HCC patients were calculated using the previously described formula, and the patients were classified into high- and low-risk groups, according to the median. The Kaplan–Meier plotter was utilised to estimate the difference in survival time between the high-and low-risk patients. A $p$-value $< 0.05$ for the log-rank test on both sides indicated a significant difference in survival time between the two groups.

## Construction of nomogram

We used the mutual clinical traits in the discovery and validation cohorts to construct a nomogram, using the "rms" R package. We assessed the prognostic accuracy of the nomograms by evaluating the corresponding calibration plots. The predicted and observed results of the nomogram are shown in a calibration curve; the 45° line indicated the best prediction.

## Statistical analysis

A time-dependent receiver operating characteristic (ROC) analysis was performed to explore the prognostic accuracy of the classifier, based on multiple differentially expressed MRGs, using the "survivalROC" package in R. The Kaplan–Meier plotter was used to analyse the OS, RFS, and PFS of the discovery cohort, and the OS of the validation cohort. Statistical differences between the groups were evaluated using the log-rank test. Univariate and multivariate Cox regression analyses were performed to assess the prognostic value of clinical characteristics and the risk score, and clinical stratified analyses were performed to test whether the risk score was independent of other clinical features, including age, sex, risk factors (HBV + HCV, alcohol intake), degree of fibrosis (no fibrosis or fibrosis and cirrhosis), tumour grade (G), American Joint Committee on Cancer (AJCC) stage, and tumour (T) stage, which were used as covariates. Hazard ratios (HRs) and their respective 95% confidence intervals (CI) were obtained. A $p$-value $< 0.05$ was considered statistically significant. The statistical tests were bilateral and conducted using R software (version 3.5.3)

## Gene set enrichment analysis (GSEA)

Based on the median risk, 343 HCC samples from TCGA group and 230 HCC samples from the ICGC group were divided into high- and low-risk groups. To identify metabolic and other important changes in the Kyoto Gene and Genome Encyclopaedia (KEGG) pathways, we used GSEA version 4.0.1 to perform a GSEA between the high- and low-risk populations. The annotated gene set (c2.cp.kegg.v7.0.symbols.gmt) was defined as the reference gene set. False discovery rate (FDR) $< 0.05$ was set as the cut-off.
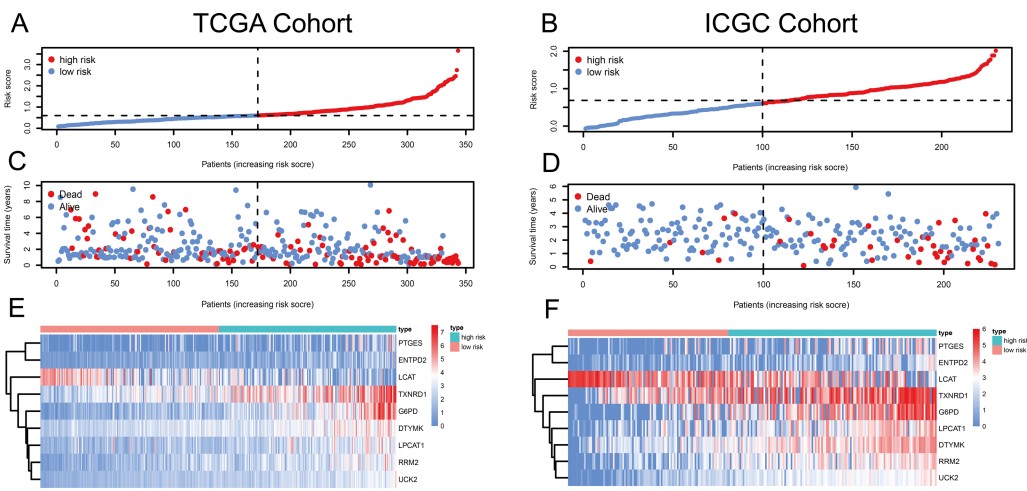

**Figure 1** Risk score, reflecting overall survival, based on the metabolism-related-gene signature comprising nine genes, in the TCGA and ICGC cohorts. (A–B) Risk-score distribution in the TCGA and ICGC cohorts. (C–D) The survival status of patients in the high- and low-risk groups in the TCGA and ICGC cohorts. (E–F) Heatmap of the expression of the nine MRGs in the high- and low-risk groups and the TCGA and ICGC cohorts.

## RESULTS

### Differential MRG expression

MRG matrices, consisting of 50 normal and 374 HCC samples from TCGA and 202 normal and 243 HCC samples from the ICGC, were obtained. Nine downregulated and 45 upregulated MRGs were identified. Univariate Cox regression analysis was used to identify prognostic MRGs.

### Identification of the signature based on multiple MRGs

Using univariate Cox regression analysis, 22 MRGs closely associated with OS were identified (Table S1). To prevent over-fitting of MRGs and to determine the best gene combination for the signature, we performed LASSO regression analysis. Finally, nine MRGs were identified to establish a prognostic model. We used the coefficients of the multivariate Cox regression model, which combined the predictions of the expression of the nine MRGs and their corresponding survival times and survival states to construct the following risk score formula: risk score = $(0.0193 \times RRM2$ expression) $+ (0.0068 \times DTYMK$ expression) $+ (0.0003 \times LPCAT1$ expression) $+ (−0.0013 \times LCAT$ expression) $+ (0.0087 \times TXNRD1$ expression) $+ (0.0035 \times G6PD$ expression) $+ (0.0012 \times PTGES$ expression) $+ (0.0508 \times ENTPD2$ expression) $+ (0.0729 \times UCK2$ expression) (Fig. 1). Among the nine prognostic MRGs, *LCAT* had a negative coefficient in Cox regression analysis, suggesting that its higher expression level was associated with lower risk and better OS. In contrast, *RRM2*, *DTYMK*, *LPCAT1*, *TXNRD1*, *G6PD*, *PTGES*, *ENTPD2*, and *UCK2* showed positive coefficients; their elevated expression levels were accompanied by higher risk scores, thus predicting poor OS. According to the median risk score of 0.60, patients were divided into high- and low-risk groups. A risk score greater than 0.60 was considered

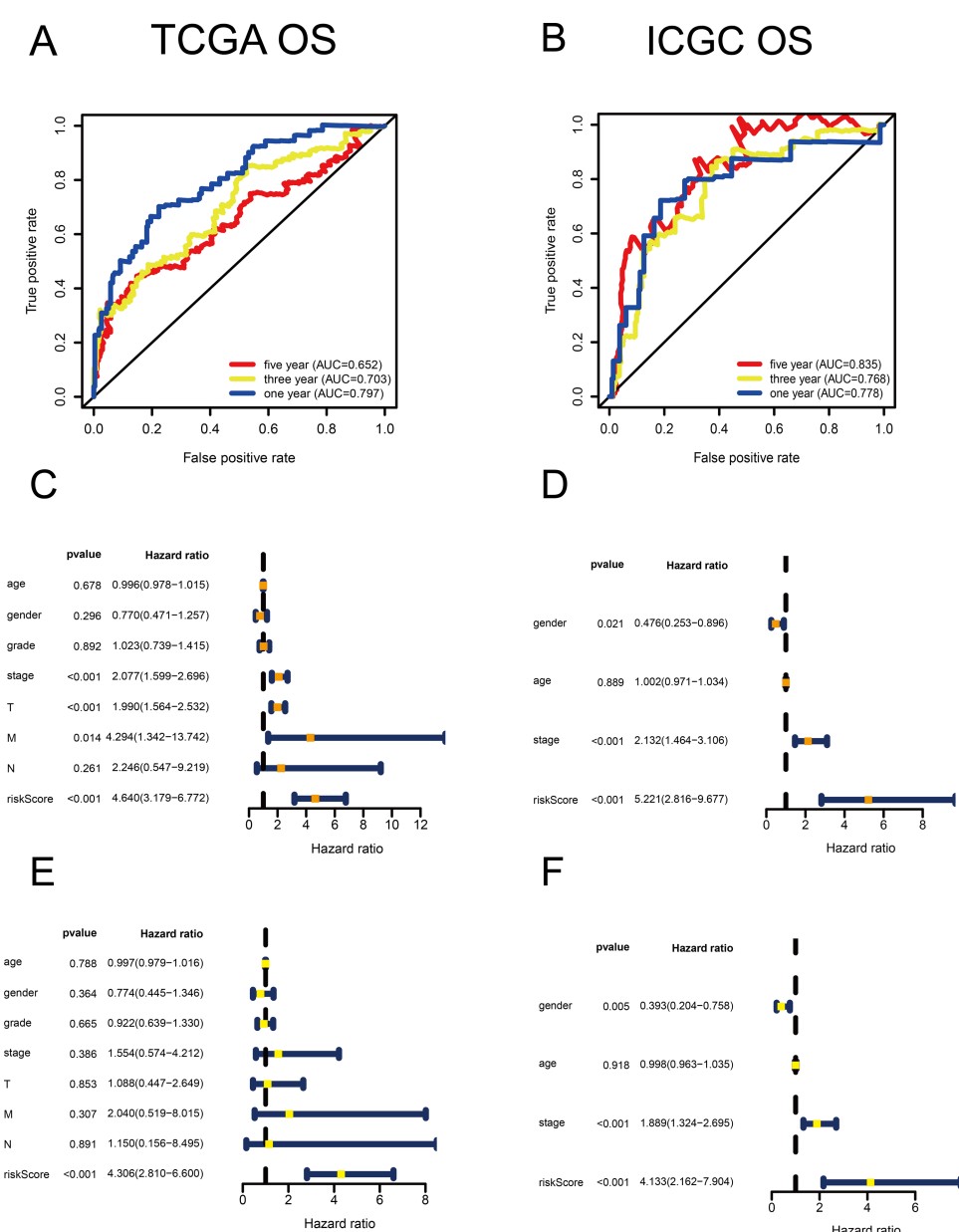

**Figure 2** **Validation of the risk scores based on the metabolism-related gene (MRG) signature comprising nine genes.** (A–B) Time-dependent receiver operating characteristic (ROC) curves of the MRG signature in the TCGA and ICGC cohorts. (C–D) Univariate Cox regression analysis for TCGA and ICGC cohorts. (E–F) Multivariate Cox regression analysis for TCGA and ICGC cohorts.

to indicate high risk, whereas a risk score $\leq 0.60$ was considered to indicate low risk. The accuracy of the OS predictions, based on the MRG signature, was evaluated by generating the ROC curves over time at 1, 3, and 5 years. A higher area under curve (AUC) represents a better prognostic capability. The AUCs based on the MRG signatures in the discovery and validation cohorts were 0.797, 0.703, and 0.652 (in TCGA cohort), and 0.778, 0.768, and 0.835 (in the ICGC cohort), respectively (Figs. 2A and 2B). These results suggested that

the prognostic model had high sensitivity and specificity. The results of the univariate and multivariate Cox analyses showed that the risk score remained an independent predictor, after adjusting for other covariates, in both the discovery and validation cohorts, and that gender and stage acted as independent predictors in the validation cohort (Figs. 2C–2F).

## Prognostic value of the MRG signature

To investigate the association between the MRG signature and prognosis, we performed an analysis of OS, RFS, and PFS in the high- and low-risk groups of the discovery cohort (Table 1), and that of OS in the high- and low-risk groups of the validation cohort (Table 2, Fig. 3). In the discovery cohort, patients in the low-risk group had longer OS (HR = 2.20, 95% CI [1.51–3.19], $p < 0.001$), RFS (HR = 1.62, 95% CI [1.16–2.27], $p = 0.005$), and PFS (HR = 1.64, 95% CI [1.21–2.20], $p = 0.001$) than patients in the high-risk group. We also used the validation cohort to verify OS (HR = 5.22, 95% CI [2.81–9.68], $p < 0.001$). As predicted, patients in the high-risk group had higher mortality than patients in the low-risk group. Thus, the MRG signature can be considered an indicator of risk for HCC.

## Associations between the MRG signature and OS by sex and age

To investigate the impact of clinical characteristics on the prognostic value of the MRG signature, we categorised HCC patients based on six clinical traits: age, sex, risk factors (HBV + HCV, alcohol intake), tumour grade (G), AJCC stage, and tumour (T) stage. This stratified analysis revealed that OS, RFS, and PFS were significantly affected in male patients and those aged ≤ 65 years in the discovery cohort; it also revealed significant OS results in the validation cohort (Figs. 4 and 5). Moreover, the OS, RFS, and PFS results for patients in the alcohol-intake, degree of fibrosis (no fibrosis or fibrosis and cirrhosis), G1–G2, and T1–T2 groups in the discovery cohort were also significant . However, the corresponding clinical information was not complete for the validation cohort. The prognostic stratified analysis showed that the survival rate of male patients in the low-risk group was significantly higher (OS [HR = 2.85, 95% CI [1.73–4.71], $p < 0.001$], RFS [HR = 1.77, 95% CI [1.18–2.66], $p = 0.006$], and PFS [HR = 1.76, 95% CI [1.21–2.55], $p = 0.003$]) than that of patients in the high-risk (validation cohort) group (OS [HR = 4.52, 95% CI [1.66–12.25], $p = 0.003$]). In addition, the survival rate of patients aged ≤ 65 years in the low-risk group was significantly higher (OS [HR = 1.97, 95% CI [1.21–3.21], $p = 0.007$], RFS [HR = 1.62, 95% CI [1.06–2.47], $p = 0.027$], and PFS [HR = 1.64, 95% CI [1.13–2.39], $p = 0.009$]) than that of patients in the high-risk (ICGC) group (OS [HR = 5.58, 95% CI [1.55–20.11], $p = 0.009$]).

## Associations between the MRG signature and OS in patients carrying wild-type *TP53* or *CTNNB1*

The tumour suppressor gene *TP53* and the oncogene *CTNNB1* are most commonly mutated in HCC and are associated with poor prognosis. Numerous studies, including epigenetic studies, have shown that mutations are important drivers of tumour carcinogenesis. Therefore, we performed a stratified analysis based on the mutation status of *TP53* or *CTNNB1* (Fig. 6). For patients with wild-type *TP53* or *CTNNB1*, the probability of death

Tang et al. (2020), *PeerJ*, DOI 10.7717/peerj.9774

**Table 1** Association between survival probability and the metabolism-related-gene signature comprising nine genes, in hepatocellular carcinoma patients in the TCGA cohort.

| Parameter | OS cohort | | | RFS cohort | | | PFS cohort | | |
|---|---|---|---|---|---|---|---|---|---|
| | Volume (High/Low) | HR (95% CI) | *P*-value | Volume (High/Low) | HR (95% CI) | *P*-value | Volume (High/Low) | HR (95% CI) | *P*-value |
| Total | 171/172 | 2.20 (1.51–3.19) | <0.001 | 150/151 | 1.62 (1.16–2.27) | 0.005 | 169/168 | 1.64 (1.21–2.20) | 0.001 |
| Age | | | | | | | | | |
| ≤ 65 | 108/108 | 1.97 (1.21–3.21) | 0.007 | 97/94 | 1.62 (1.06–2.47) | 0.027 | 108/104 | 1.64 (1.13–2.39) | 0.009 |
| >65 | 63/64 | 2.60 (1.42–4.73) | 0.002 | 53/57 | 1.61 (0.93–2.77) | 0.088 | 61/64 | 1.62 (0.98–2.67) | 0.061 |
| Gender | | | | | | | | | |
| Male | 118/115 | 2.85 (1.73–4.71) | <0.001 | 104/105 | 1.77 (1.18–2.66) | 0.006 | 116/113 | 1.76 (1.21–2.55) | 0.003 |
| Female | 53/57 | 1.77 (0.96–3.27) | 0.069 | 46/46 | 1.34 (0.74–2.41) | 0.334 | 53/55 | 1.42 (0.85–2.38) | 0.181 |
| Risk factors | | | | | | | | | |
| HBV + HCV | 67/75 | 2.24 (1.08–4.69) | 0.031 | 59/65 | 1.08 (0.62–1.88) | 0.792 | 69/72 | 1.32 (0.83–2.10) | 0.242 |
| Alcohol intake | 56/55 | 1.93 (1.00–3.71) | 0.049 | 53/47 | 2.15 (1.17–3.95) | 0.014 | 57/53 | 1.84 (1.08–3.14) | 0.025 |
| Degree of fibrosis | | | | | | | | | |
| No fibrosis | 30/42 | 5.89(2.82–12.28) | <0.001 | 25/36 | 3.67(1.56–8.66) | 0.003 | 29/42 | 2.45 (1.34–4.50) | 0.004 |
| Fibrosis and cirrhosis | 60/67 | 3.55 (1.74–7.27) | <0.001 | 53/58 | 2.50 (1.18–5.30) | 0.017 | 61/63 | 2.55 (1.40–4.64) | 0.002 |
| Grade | | | | | | | | | |
| G1–G2 | 83/131 | 2.20 (1.38–3.49) | <0.001 | 78/113 | 1.78 (1.16–2.74) | 0.008 | 81/128 | 1.67 (1.13–2.46) | 0.010 |
| G3–G4 | 86/38 | 2.42 (1.12–5.24) | 0.025 | 70/35 | 1.54 (0.83–2.87) | 0.170 | 86/37 | 1.68 (0.96–2.95) | 0.071 |
| T classification | | | | | | | | | |
| T1–T2 | 120/132 | 2.44 (1.45–4.11) | <0.001 | 103/116 | 1.81 (1.17–2.77) | 0.007 | 120/128 | 1.94 (1.34–2.81) | <0.001 |
| T3–T4 | 51/37 | 1.90 (1.08–3.35) | 0.025 | 47/32 | 1.21 (0.70–2.10) | 0.488 | 49/37 | 1.06 (0.63–1.78) | 0.823 |
| AJCC stage | | | | | | | | | |
| Stage I | 63/98 | 2.24 (1.13–4.41) | 0.020 | 52/86 | 1.43 (0.80–2.58) | 0.230 | 62/96 | 1.53 (0.92–2.53) | 0.099 |
| Stage II | 47/30 | 2.55 (0.94–6.95) | 0.066 | 42/27 | 2.05 (0.98–4.31) | 0.057 | 47/28 | 2.09 (1.07–4.09) | 0.031 |
| Stage III and IV | 49/34 | 1.86 (1.02–3.39) | 0.043 | 45/29 | 1.21 (0.68–2.14) | 0.513 | 49/34 | 1.04 (0.61–1.78) | 0.881 |

**Notes.**

HR, hazard ratio; CI, confidence interval; HBV, hepatitis B virus; HCV, hepatitis B virus; G, grade; T, tumor; AJCC, American Joint Committee on Cancer.

**Table 2** Association between overall survival and the metabolism-related-gene signature comprising nine genes, in hepatocellular carcinoma patients in the ICGC cohort.

| Parameter | ICGC cohort | | |
|---|---|---|---|
| | Number (High/Low) | HR (95% CI) | *P*-value |
| Total | 130/100 | 5.22 (2.82–9.68) | <0.001 |
| Age | | | |
| ≤65 | 45/44 | 5.58 (1.55–20.11) | 0.009 |
| >65 | 85/56 | 6.47 (1.93–21.72) | 0.002 |
| Gender | | | |
| Male | 93/76 | 4.51 (1.66–12.25) | 0.003 |
| Female | 37/24 | 12.41 (1.64–94.14) | 0.015 |
| LCSGJ stage | | | |
| Stage I | 22/15 | 1.00 (0.82–1.22) | 0.999 |
| Stage II | 59/41 | 2.68 (0.98–7.36) | 0.056 |
| Stage III and IV | 51/45 | 17.68 (2.31–135.11) | 0.006 |

**Notes.**
HR, hazard ratio; CI, confidence interval; LCSGJ, Liver Cancer Study Group of Japan.

was significantly higher in the high-risk group than in the low-risk group. These results were consistent between the data from the two independent databases, TCGA and ICGC. Therefore, this MRG signature can be considered a risk indicator for patients carrying wild-type *TP53* or *CTNNB1*.

## Independent data set drug trial validation using the GEO database

To explore the association between the MRGs and drug response, we analysed differences in expression of the MRGs, using the GSE109211 dataset from the GEO database. This dataset comprised 67 patients treated with sorafenib (46 non-responders, 21 responders) and 73 patients who received placebo treatment (Fig. 7). Compared with that in the placebo and sorafenib non-responder groups, the expression of *TXNRD1*, *LCAT*, and *G6PD* in the sorafenib responder group showed significant downregulation, and that of *PTGES*, *RRM2*, and *ENTPD2* showed significant upregulation. The expression of *UCK2* was statistically significant in the sorafenib non-responder and responder groups, and was downregulated in the sorafenib responder group, relative to the expression in the non-responder and placebo groups. The expression levels of *LPCAT1* and *DTYMK* were not significantly different between patients treated with sorafenib (responders and non-responders) and those treated with the placebo. In other words, the MRGs in the model showed a good response to sorafenib, which suggests the effectiveness of our MRG signature in predicting the prognosis of patients with HCC.

## Nine MRGs for GSEA

GSEA was conducted to ascertain significant changes in the potential pathways between high-risk and low-risk populations. Based on the selection criteria of an FDR < 0.05, four significantly altered pathways in the high-risk group were observed: the p53 signalling pathway, the cell cycle pathway, purine metabolism, and pyrimidine metabolism (Fig. 8).

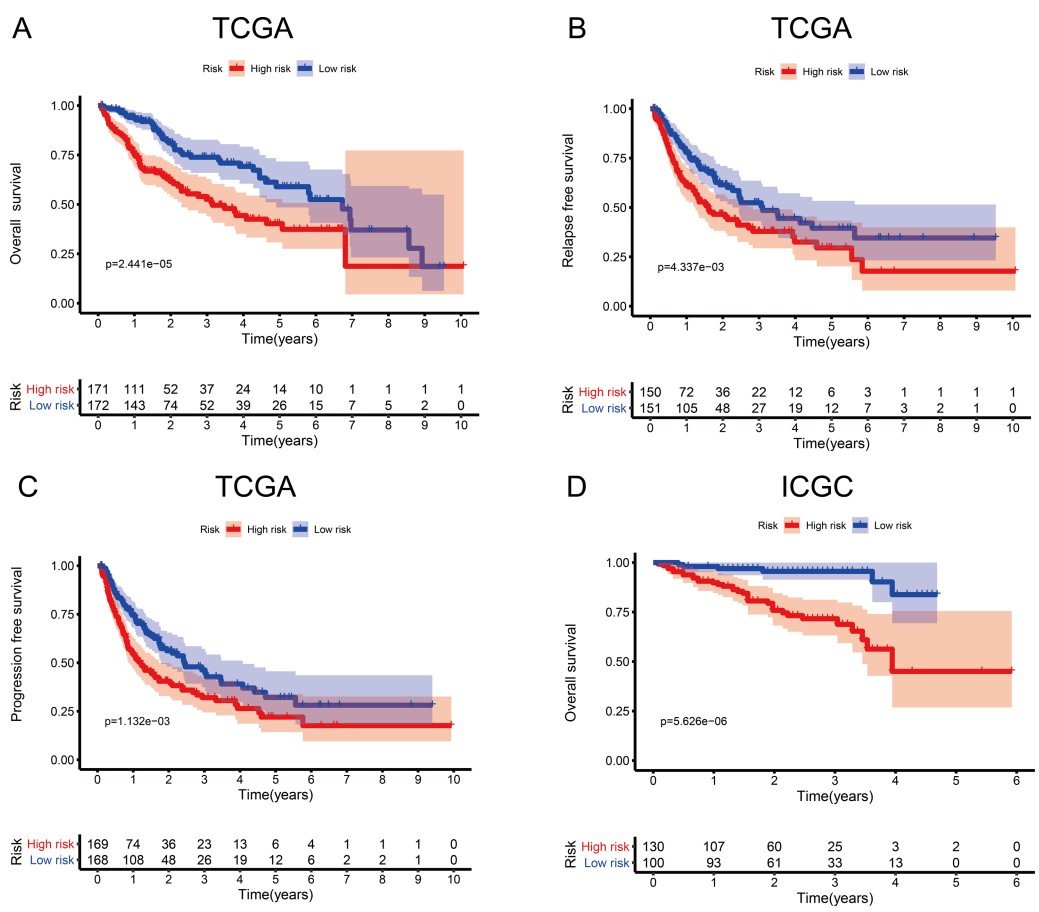

**Figure 3  Survival probability based on the metabolism-related gene (MRG) signature comprising nine genes.** (A) Overall survival, OS; (B) relapse-free survival, RFS; and (C) progression-free survival, PFS, in the TCGA cohort. (D) OS in the ICGC cohort.

## Construction of nomograms

A quantitative method was used, and a nomogram was constructed to predict the 1-, 3-, and 5-year OS of patients with HCC in the discovery and validation cohorts, based on their common clinical traits (Fig. 9). The calibration curves revealed that the nomograms for the discovery and the validation cohorts had good accuracy, as observed in the ideal model.

## DISCUSSION

HCC accounts for more than 80% of liver cancer cases. It is a highly malignant, recurrent, and drug-resistant cancer that is often diagnosed at an advanced stage. Metabolic changes are widely reported characteristics of HCC (*Grandhi et al., 2016*). Although the three major liver metabolism pathways (glucose, lipid, and protein pathways) have been identified, these are far from enough to reveal the metabolic changes related to HCC (*De Matteis et al., 2018*; *Lee et al., 2016*; *Liang et al., 2018*; *Shang, Qu & Wang, 2016*; *Tang et al., 2018*). We constructed an MRG signature comprising nine genes, using TCGA data, and further verified its association with OS using an independent dataset to confirm the sensitivity

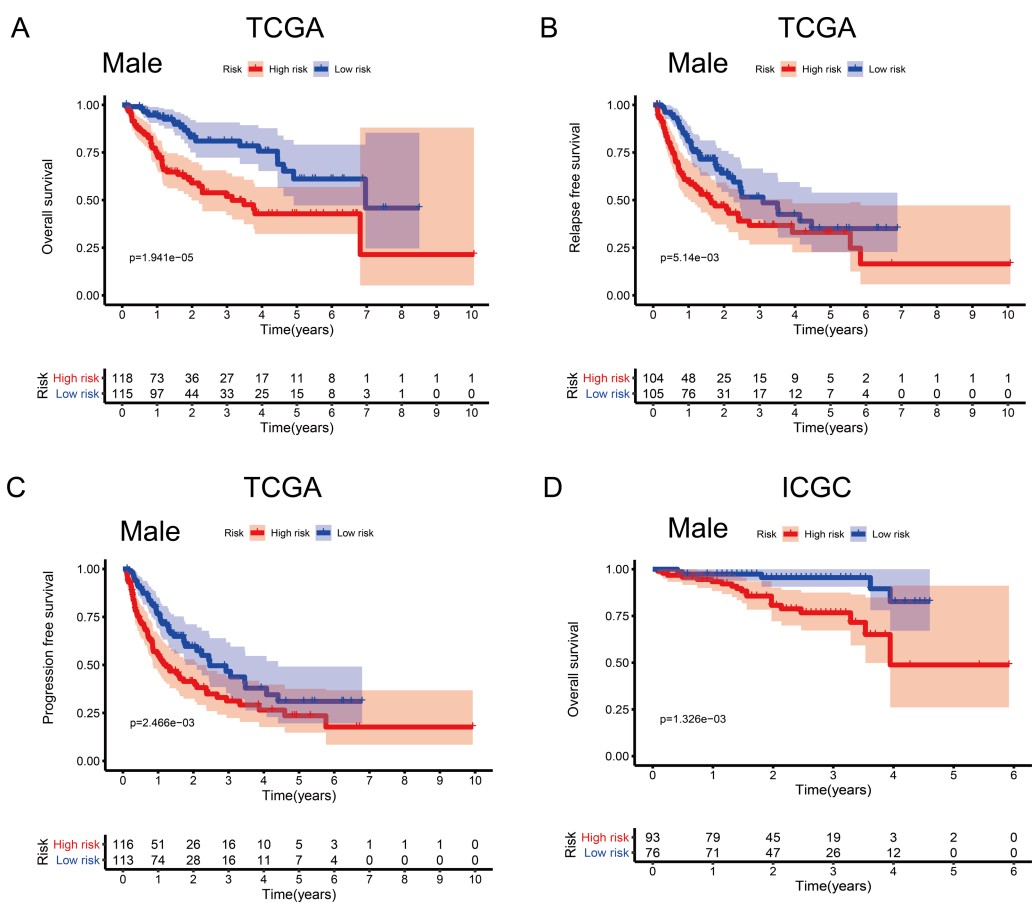

**Figure 4** **Association between survival and the metabolism-related gene (MRG) signature comprising nine genes, in male hepatocellular carcinoma patients.** (A) Overall survival, OS; (B) relapse-free survival, RFS; and (C) progression-free survival, PFS, in the TCGA cohort. (D) OS in the ICGC cohort.

and specificity of the model. The MRG signature had statistically significant prognostic value for male patients, those ≤ 65 years, and those carrying wild-type *TP53* or *CTNNB1*. Next, we performed independent drug verification of this signature using a GEO dataset, revealing that some of the identified genes showed good response to sorafenib. Overall, these results demonstrate the effectiveness of the MRG signature.

Clinical findings have shown that the incidence of HCC and the associated mortality rates are higher in men than in women (*Sukocheva, 2018*). This has been attributed to the lack of a protective effect of high oestrogen levels in men who drink heavily (*Baecker et al., 2018*; *Montano-Loza et al., 2018*; *Singh et al., 2019*). Although the pathogenesis of alcohol-induced HCC is complicated and still unclear, it is certain that alcohol toxicity causes liver cells to catabolise fatty acids, resulting in fat accumulation and fibrosis. Additionally, alcohol inhibits natural killer cells, which play key roles in antiviral, antitumour, and antifibrotic defence in terms of innate immunity. It also impairs the proteasome functions of macrophages and dendritic cells in terms of adaptive immunity, thus altering the presentation of alloantigens. Furthermore, the oxidative metabolites of alcohol can

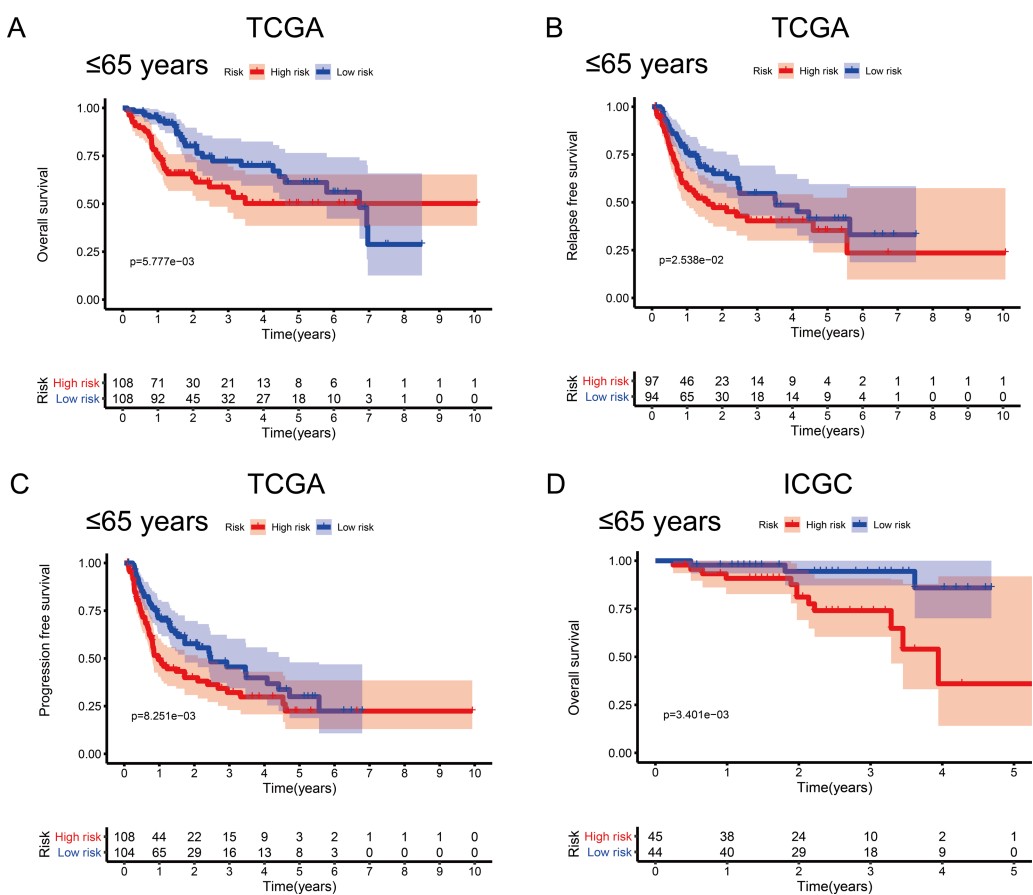

**Figure 5** Association between survival probability and the metabolism-related gene (MRG) signature comprising nine genes, in hepatocellular carcinoma patients aged $\leq$ 65 years. (A) Overall survival, OS; (B) relapse-free survival, RFS; and (C) progression-free survival, PFS, in the TCGA cohort. (D) OS in the ICGC cohort.

interfere with DNA methylation, synthesis, and repair, promote HCC carcinogenesis, and increase HCC sensitivity (*Ceni, Mello & Galli, 2014*; *Miller et al., 2011*). In our study, we observed that in TCGA cohort, alcohol intake was significantly greater in patients aged $\leq$ 65 years than in patients aged > 65 years, and the drinking history was longer in the former. Furthermore, mutant alleles for *TP53* and *CTNNB1* were more frequent in men (33.5% and 34.7%, respectively) than in women (21.8% and 12.7%, respectively). The MRG signature had a significant prognostic value for patients carrying wild-type TP53 and CTNNB1 in both cohorts and also mutant TP53 and CTNNB1 in the ICGC cohort (Fig. S1), thus suggesting that *TP53* and *CTNNB1* are risk-indicator genes for HCC, regardless of the presence or absence of mutations. More importantly, seven genes in the model showed good responsiveness to sorafenib. Studies by *Lee et al. (2019)* showed that thioredoxin reductase 1 (*TXNRD1*) is significantly overexpressed in cytoplasmic subunits and is a key enzyme of the thioredoxin system, and is related to poor clinical pathological features and survival outcomes of HCC patients. Targeting *TXNRD1* results in accumulation of reactive

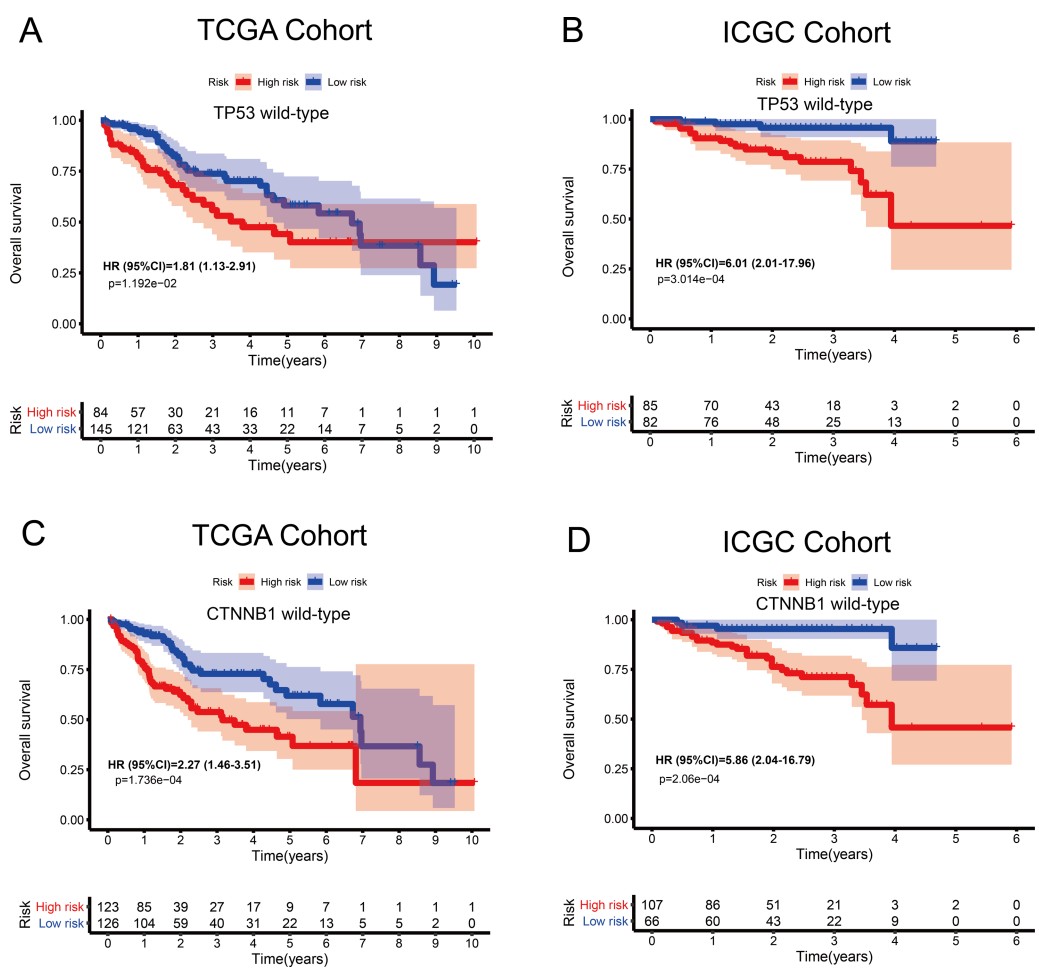

**Figure 6** Kaplan–Meier curves depicting overall survival (OS) in hepatocellular carcinoma patients. (A–B) Patients with wild-type *TP53* in the TCGA and ICGC cohorts, respectively. (C–D) Patients with wild-type *CTNNB1* in the TCGA and ICGC cohorts, respectively.

oxygen species, causing cancer cells to experience high levels of oxidative stress. Auranofin, a *TXNRD1* inhibitor, effectively exacerbates oxidative stress. This is consistent with our validation of sorafenib sensitivity using the GEO data set, which revealed that *TXNRD1* expression was significantly downregulated in the sorafenib responder group relative to that in the placebo and sorafenib non-responder groups. Considering the relatively small sample size in this study, the results must be validated in larger cohorts.

We used the common clinical traits of patients in TCGA and ICGC groups to construct a new nomogram, based on the expression of the nine MRGs, to accurately predict 1-, 3-, and 5-year survival rates. However, our sample sizes were limited, and only patients with complete information were included in the analysis, thereby limiting the precision of our estimates. However, we found that five (RRM2, TXNRD1, DTYMK, UCK2 and ENTPD2 ) of the nine MRGs were enriched in the purine metabolism and pyrimidine metabolism pathways, which was an important finding and indicated that the purine

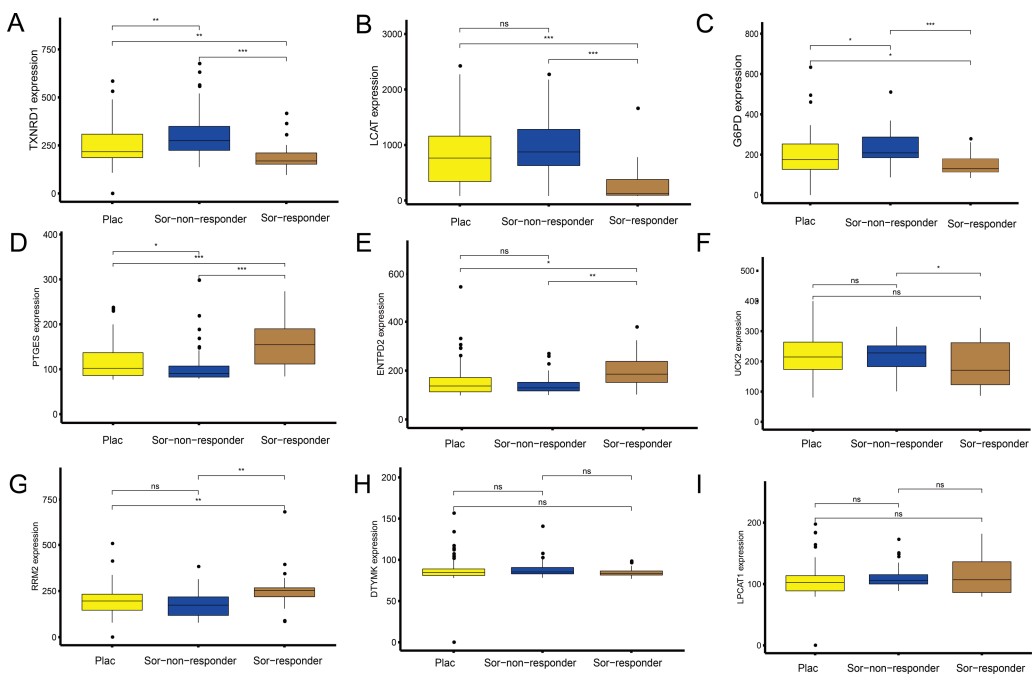

**Figure 7  Sorafenib sensitivity validation for the nine metabolism-related genes (MRGs) in the independent cohort from the GEO database.** (A–I) Differences in the expression of these genes, between patients who received the placebo and sorafenib treatment, and between responders and non-responders. Wilcoxon test: ***$P < 0.001$, **$P < 0.01$, *$P < 0.05$, ns, not significant.

and pyrimidine metabolism pathways had a significant impact on the prognosis of HCC. Human ribonucleotide reductase (RR) comprised of RRM1 and RRM2 can maintain the steady state of the nucleotide library by converting ribonucleoside diphosphate to 2′-deoxyribonucleoside diphosphate, and increased expression and activity of RR are related to malignant transformation and growth; therefore, the key role of RR in DNA synthesis and repair makes it an important anticancer target. For example, the anti-RRM2 siRNA duplex shows anti-proliferative activity in cancer cells. In addition, in vitro experiments verified that sorafenib inhibited the expression of RRM2 in HCC cells, which was positively correlated with the anticancer activity of sorafenib, proving that RRM2 is a new molecular target of sorafenib in HCC cells (*Yang, Lin & Liu, 2020*). Ectonucleoside triphosphate diphosphohydrolase 2 (ENTPD2) can be induced by hypoxia by stabilising hypoxia-inducible factor 1 (HIF-1) and its overexpression in clinical specimens of HCC. Myeloid-derived suppressor cells (MDSC) have immunosuppressive activity, which can enable cancer to evade immune surveillance and become unresponsive to immune checkpoint blockade. Hypoxia is the cause of MDSC accumulation. ENTPD2 converts extracellular ATP to 5′-AMP, which prevents the differentiation of MDSC and promotes their maintenance. In this way, MDSC can promote tumour immune escape (*Chiu et al., 2017*). Uridine-cytidine kinase 2 (UCK2) is a pyrimidine ribonucleoside kinase that catalyses the phosphorylation of uridine and cytidine to UMP and CMP. The enzyme also catalyses the phosphorylation of several cytotoxic ribonucleoside analogues, and it has

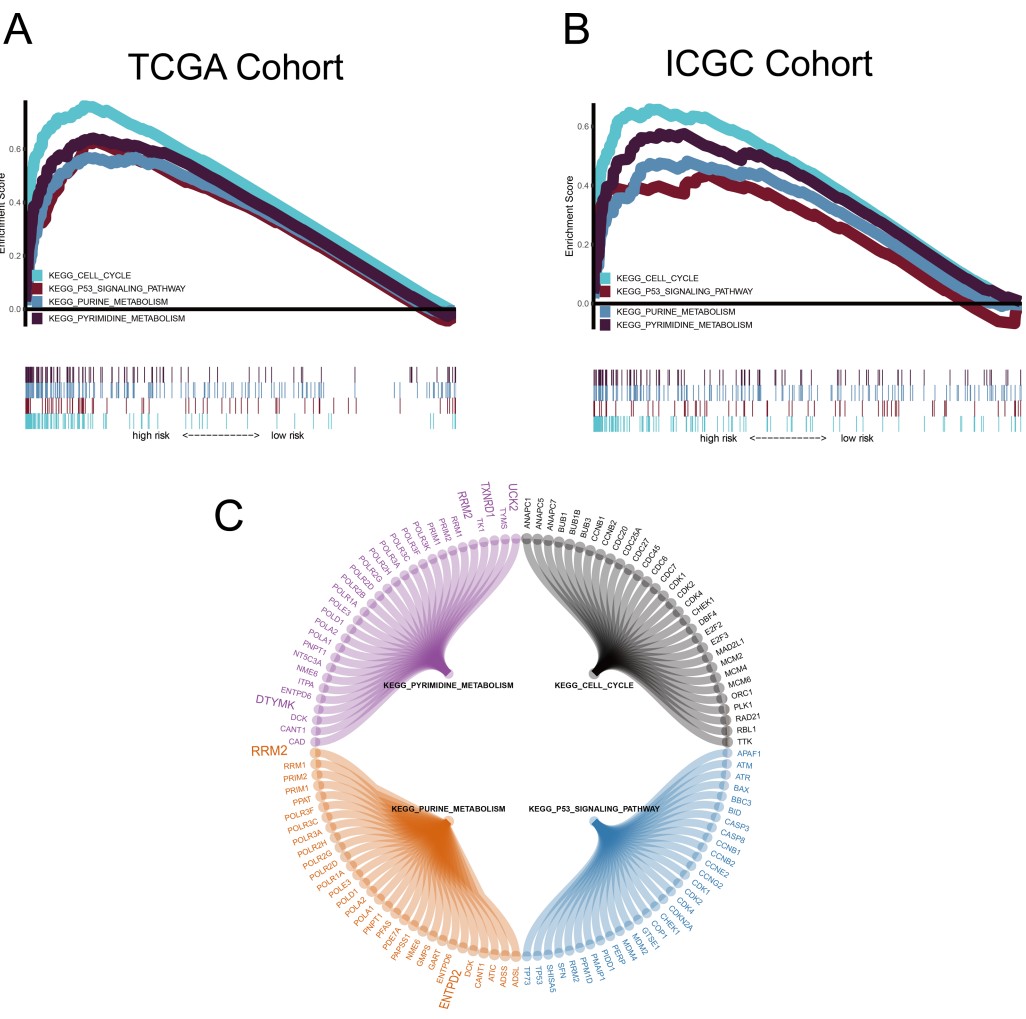

**Figure 8 Gene set enrichment analysis (GSEA).** (A–B) Metabolism-related gene (MRG) signature comprising the nine genes that had a falsediscovery rate < 0.05. (C) Circle diagram showing the gene components of four pathways.The large genes are those used in the MRG signature.

been studied as a possible chemotherapeutic agent for the treatment of cancer (*Malami & Abdul, 2019*; *Murata et al., 2004*). In vitro experiments show that UCK2 knockdown can inhibit cell migration, invasion, and proliferation, whereas overexpression of UCK2 has the opposite effect. Animal model experiments confirmed that knocking out UCK2 can inhibit tumour growth in vivo (*Huang et al., 2019*). However, the specific underlying mechanism needs to be further verified. As an important gene that controls dTTP biosynthesis and DNA replication, deoxythymidylate kinase (*DTYMK*) is necessary for all dividing cells. Studies have shown that DTYMK knockdown and LKB1 loss DTYMK are synthetically lethal, that is, excessive consumption of DTYMK below a critical threshold is lethal to all dividing cells, especially those carrying low levels of deoxynucleotide pools, such as tumour cells that maintain a rapid growth rate (*Liu et al., 2013*). LKB1/STK11 is the main regulator of cell metabolism and energy stress response. The best characterised target is

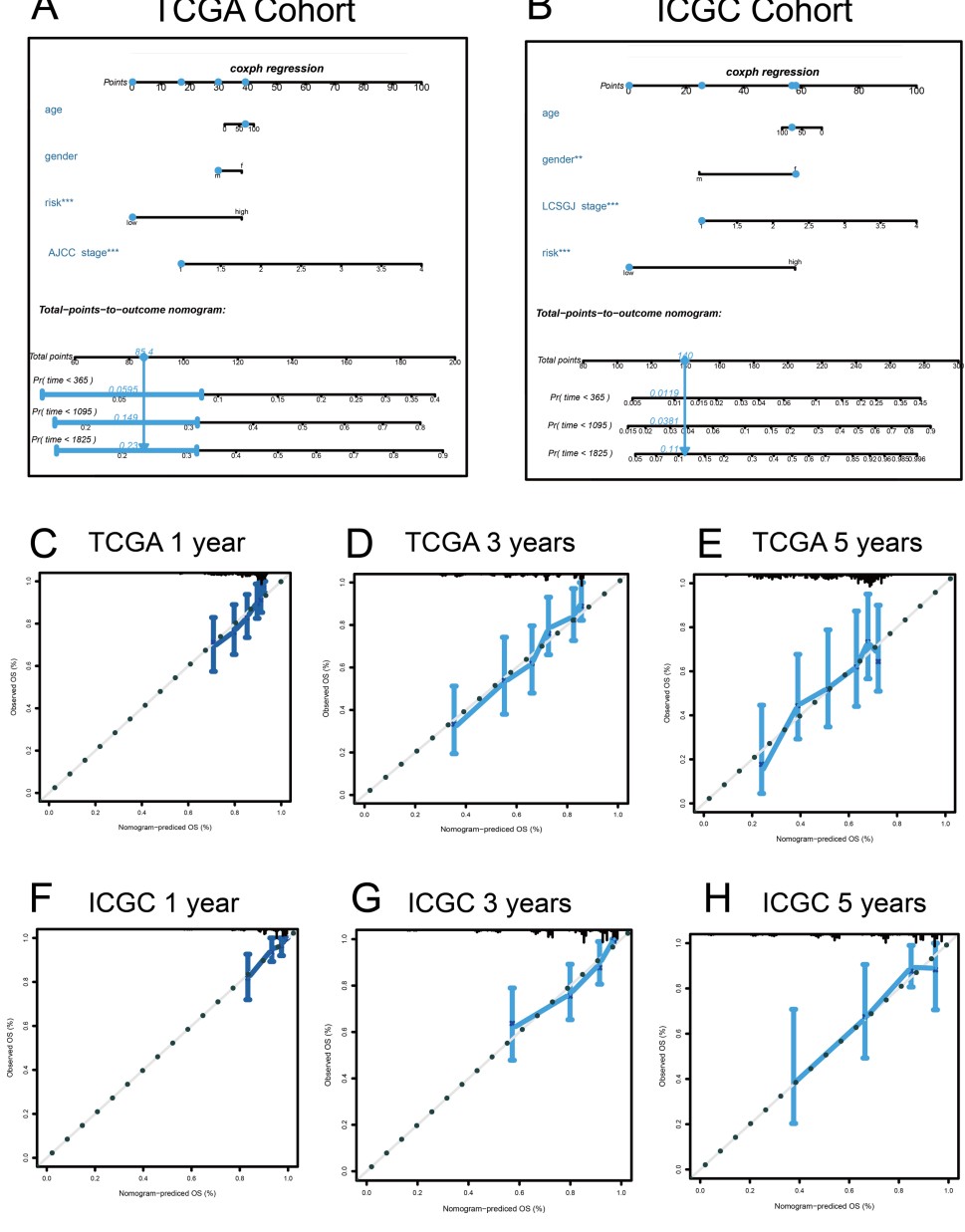

**Figure 9 Nomograms to predict overall survival (OS) in hepatocellular carcinoma patients.** (A–B) Nomograms using clinical traits shared between the TCGA and ICGC cohorts. Nomogram calibration curves for 1-, 3-, and 5-year OS, in the TCGA (C–E) and ICGC (F–H) cohorts. AJCC, American Joint Committee on Cancer; LCSGJ, Liver Cancer Study Group of Japan.

AMP-activated protein kinase (AMPK). In the case of low cellular ATP levels, AMPK is activated and directly phosphorylated by LKB1. AMPK in turn regulates the use of nutrients through the phosphorylation of various substrates, thereby restoring energy homeostasis and controlling the absorption and metabolism of nutrients (*Mihaylova & Shaw, 2011*). LKB1/STK11 deficiency can lead to extensive defects in metabolic control.

Primary cells and cancer cell lines lacking LKB1 are sensitive to nutritional deprivation and other types of metabolic stress as evidenced by this (*Wingo et al., 2009*). This is a direct evidence that DTYMK affects the occurrence and development of cancer through LKB1. Notably, the enrichment results of RRM2, DTYMK, UCK2, and ENTPD2 in GSEA were highly consistent with their corresponding KEGG pathways. TXNRD1 was enriched in the pyrimidine metabolism pathway in GSEA, but has not yet been included in the pyrimidine metabolism pathway in the KEGG database (Fig. 8C and Fig. S2). There may be potential mechanisms that have not yet been discovered, and this important observation may have been ignored by previous researchers. The risk score of our model was proved to be an independent prognostic factor of HCC and it allows accurate evaluation of HCC prognosis.

## CONCLUSIONS

In summary, we have systematically demonstrated the prognostic value of our MRG signature for male patients, patients aged $\leq$ 65 years, and patients carrying the wild-type *TP53* and *CTNNB1* genes. We have revealed the association between this gene signature and sorafenib responsiveness and explained the causes of the different responses among the groups. These findings provide new information regarding HCC prevention, diagnosis, and prognosis, and can be used in developing precision medicine approaches for individualised treatment.

## ACKNOWLEDGEMENTS

We would like to thank Editage for English language editing.

### Funding
This work was supported by the National Natural Science Foundation of China (2016GXNSFAA380306) and the self-generated project of the Guangxi Zhuang Autonomous Region Health Department (Z20170816), China. The funders had no role in study design, data collection and analysis, decision to publish, or preparation of the manuscript.

### Grant Disclosures
The following grant information was disclosed by the authors:
National Natural Science Foundation of China: 2016GXNSFAA380306.
Guangxi Zhuang Autonomous Region Health Department: Z20170816.

### Competing Interests
The authors declare there are no competing interests.

## Author Contributions

- Chaozhi Tang and Jiakang Ma conceived and designed the experiments, performed the experiments, analyzed the data, prepared figures and/or tables, authored or reviewed drafts of the paper, and approved the final draft.
- Xiuli Liu and Zhengchun Liu conceived and designed the experiments, analyzed the data, authored or reviewed drafts of the paper, and approved the final draft.

## Data Availability

The data is available at the Cancer Genome Altas (TCGA, https://cancergenome.nih.gov). under TCGA search term: TCGA-LIHC and from the International cancer genomics consortium (ICGC, https://icgc.org) using ICGC search term: LIRI-JP (https://dcc.icgc.org/releases/current/Projects/LIRI-JP), and NCBI GEO: GSE109211.

## Supplemental Information

Supplemental information for this article can be found online at http://dx.doi.org/10.7717/peerj.9774#supplemental-information.

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
