# Peer review of "Identification of a prognostic signature of nine metabolism-related genes for hepatocellular carcinoma"

_PeerJ, doi:10.7717/peerj.9774_

## Round 0.1 · original submission · Major Revisions

The authors need to clarify/discuss why they chose to analyze MRGs. They also need to discuss the novelty of their 9 candidate prognostic genes based on previous literature along with all of the concerns raised by the reviewers.

Reviewer 1 ·

Basic reporting

In this study, the authors determined a list of patient prognosis predicting differentially expressed metabolism-related genes (MRGs) signature by analysing RNA sequencing data of hepatocellular carcinoma (HCC) samples in The Cancer Genome Atlas (TCGA) and International Cancer Genome Consortium (ICGC) databases. They also determined that the identified genes signature has association with sorafenib responsiveness; and also has prognostic value for HCC patients who are male, ≤65 years old, and wild-type for TP53 and CTNNB1 genes.

The manuscript was written using clear professional English. However, sufficient background/context with literature references was not provided in the manuscript. For example, the background information and reasoning for why the authors aimed to focus on analysing the differential expression of MRGs to generate a prognostic model for HCC should have been clearly explained. However, the authors provided only one sentence without a reference. In addition, references of some of the background information were not presented; such as sentences in lines 61-62 and 64-66. Furthermore, the mutations of TP53 and CTNNB1 genes were defined as the major cancer drives for HCC in the manuscript. However the genetic etiology of HCC is more complex and cannot be explained with mutations of two genes. Therefore the reasoning for focusing on mutations of TP53 and CTNNB1 genes in the study should be peformed by providing a more approriate background information, as well.

Experimental design

The following major issues about the experimental design of the study have been determined:

• The method for generation of the list of MRGs were described in the manuscript with only one sentence as “MRGs were extracted from the file “c2.cp.kegg.v7.0.symbols.gmt” in the gene set enrichment analysis (GSEA) database.” The list generation method was described with insufficient information to replicate. Because,
i. the total number or the list of MRGs were not presented,
ii. “c2.cp.kegg.v7.0.symbols.gmt” file contains 186 different genesets. There are lots of genesets, which are irrelevant to metabolism, in this gmt file, such as KEGG_OOCYTE_MEIOSIS, KEGG_CARDIAC_MUSCLE_CONTRACTION, KEGG_AXON_GUIDANCE, etc. Whether these genesets were discarded or included in the study was not explained in the manuscript.

• GSEAs were performed using data of 343 HCC samples from TCGA database and 230 samples from ICGC database, using the same genesets list, which was used to generate the MRGs list (c2.cp.kegg.v7.0.symbols.gmt). However, the initial number of HCC samples analysed from TCGA and ICGC databases were 374 and 243, respectively. The reason of this difference was not described. In addition, it would be better to perform GSEAs by using the complete c5 and c2.cp genesets lists, since these lists include genesets from additional pathway and ontology databases, such as Gene Ontology and Biocarta. This approach might significantly improve the experimental design of the study, because it is stated in the manuscript that “Further, we were unable to discover the underlying mechanisms of some of the genes in the signature”, and analysing additional databases may provide valuable mechanistic data from a different source.

• The authors, without providing a reference, stated that “The tumor suppressor gene, TP53, and the oncogene, CTNNB1, are the two most commonly mutated genes in HCC and are associated with poor prognosis.” Although TP53 and CTNNB1 genes are the two most frequently mutated genes (31% and 27%, respectively), according to exome sequencing data of HCC samples in TCGA database there are also other known frequently mutated genes, such as NCOR1 (22%), RB1 (19%), and ALB (13%) (1). Therefore, either all known frequently mutated genes should be analysed and reported in the study or the satatement should be revised in the manuscript to provide an unbiased reason for focusing on TP53 and CTNNB1 mutations to reveal an association between MRG signature and OS in HCC patients.

References
1. Cancer Genome Atlas Research Network et al. Comprehensive and Integrative Genomic Characterization of Hepatocellular Carcinoma. Cell, 2017. 169(7):1327-1341.e23.

Validity of the findings

The following major issues about the experimental design of the study have been determined:

• Since all analyses were performed after determining the list of MRGs, the previously mentioned problems about the experimental design of the study effect all results and limit the validity of all the associated and following findings.

• All the differentially expressed MRGs lists, namely the 54 differentially expressed MRGs list, 22 differentially expressed overall survival-related genes list, and the nine MRGs list, which was used to build the prognostic model for HCC, contain genes with very low mean expression values (as seen in figure 1E and Table 1S). Although fold change values of these genes are high and p values are statistically significant, very low expression values of these genes suggest that molecules encoded from these genes may not be crucial for metabolic functions of both normal liver and HCC cells. Therefore a cut off value should be determined for identification of differentially expressed MRGs. As a result, the validities of all the following findings that rely on MRGs list are limited, because the results were obtained based on very-lowly exppressed MRGs inluding genes list.

Reviewer 2 ·

Basic reporting

Tang et al. described a prognostic gene signature for HCC, which is composed of nine metabolism related genes. The gene signature is validated in an independent dataset and seems to have some predictive power in terms of overall survival in male patients <65 years of age and patients carrying the wild-type TP53 or CTNNB1 genes. The manuscript is written in clear and technically correct English. My only criticism here is for line 269-270; `..Auranofin, a TXNRD1 inhibitor, effectively clears this stress and sensitizes HCC cells to conventional sorafenib therapy..`
The TXNRD inhibitor shall exacerbate oxidative stress, rather than clearing it to sensitize HCC cells to sorafenib therapy.
Article figures are relevant to the content of the article, appropriately described and labelled.

Experimental design

Aim of the research is to define a metabolism associated molecular signature to stratify HCC patients into high and low risk sub-groups. However, the literature in the field is not short on prognostic gene/miRNA signatures; in the case of HCC, dozens of prognostic gene signatures have already been reported. Therefore the novelty of the current gene signature is not inherently clear, especially considering that it does not predict something directly actionable (e.g. likelihood of recurrence, metastasis, sorafenib responsiveness etc.). Comparative analysis depicting the relative efficacy of this prognostic signature over others could add potential novelty to the study.
Methods are described with sufficient detail and information.

Validity of the findings

In general, validity of the findings are rigorously tested in the manuscript. All underlying data have been provided; they are robust, statistically sound and controlled. One major criticism here though is the use of overall survival as the only clinical parameter to establish validity of the gene signature in ICGC cohort. HCC differs from most cancers in that there are two biological processes to simultaneously consider. It usually arises within a liver compromised by cirrhosis, thereby requiring effective management of two disease processes. Thus, death from disease progression may arise from distinctly different causes – recurrence and/or dissemination of the HCC, de novo new HCC arising in the cirrhotic liver, and complications of progressive cirrhosis including hepatic decompensation. Thus inclusion of disease free survival into the pipeline can in principle combine all three processes noted above – recurrence and/or dissemination of HCC, de novo new HCC arising in the cirrhotic liver, and complications of progressive cirrhosis.
As a minor point, I found it a little farfetched to consider genes depicted in the signature as potential targets of sorafenib.The involvement of the majority of these genes in the pathophysiology of HCC has not been firmly established.

·

Basic reporting

The paper is poorly written, need a clearer description of the analysis. Additional editing would help make the paper easier to read and interpret.

Experimental design

The choice of data sets is not well explained – given that these are the starting point for all of the analysis this is an oversight that is not acceptable. HCC samples descriptions should be done such as which type of cells or tissues were selected for analysis.
Systems biology and bioinformatics related studies differ from each other in terms of used algorithms. But, in this paper, there is no special algorithm or method to identify nine prognostic genes. As far as I understand, 9 prognostic genes were selected from common deg. But it is acceptable but makes the method weak.
Identification of Sorafenib response to the nine gene expression profiles is a good idea but prognostic potential should be investigated via Kaplan Meier plots. Moreover, AUC values (Fig 2) are not very good, compared with other study results. It must be discussed what is good.
In fig 1, different gene expression profiles obtained from the TCGA and ICGC cohort have seen in the heat map. Please discuss why direction of gene expressions is different? The prognostic analysis is performed based on gene expression therefore the direction of gene expression (up and down-regulation) is crucial.
Why selected only two genes (CTNNB1 and TP53)? Are there other genes that show a higher mutation in this cancer? The authors make sure there are genes with the highest mutation?

Validity of the findings

This study represent novel results in terms of nine prognostic genes. I can't be sure that novelties of the genes for HCC prognosis. Unfortunately, these genes whether or not they are novel for HCC is not discussed. So, in the discussion part, nine genes related to HCC studies should be summarized as a table. The prognostic and diagnostic potentials should also be investigated from literature.

Additional comments

The paper utilizes existing gene expression data sets for HCC from two databases. Differentially expressed genes were identified and common 9 prognostic genes were found. And then further analysis are about the validation of the prognostic potential of nine genes in HCC. Unfortunately, the idea is not interesting, there are similar lots of studies.
Some of them: Comprehensive analysis of potential prognostic genes for the construction of a competing endogenous RNA regulatory network in hepatocellular carcinoma
C Yue, Y Ren, H Ge, C Liang, Y Xu, G Li… - OncoTargets and …, 2019 -

Development and validation of a three-gene prognostic signature for patients with hepatocellular carcinoma
B Li, W Feng, O Luo, T Xu, Y Cao, H Wu, D Yu… - Scientific reports, 2017 -

Systems biology and bioinformatics related studies differ from each other in terms of used algorithms. But, in this paper, there is no special algorithm or method to identify nine prognostic genes. As far as I understand, 9 prognostic genes were selected from common deg.

---

## Round 0.2 · Minor Revisions

The manuscript has significantly improved. The second reviewer's last concern should be addressed in the discussion.

Reviewer 1 ·

Basic reporting

The authors have made appropriate and acceptable changes regarding the requested revisions.

Experimental design

The authors have made appropriate and acceptable changes regarding the requested revisions.

Validity of the findings

The authors have made appropriate and acceptable changes regarding the requested revisions.

Reviewer 2 ·

Basic reporting

The authors did a fine job in improving the clarity of the text. Literature is appropriately cited; more detailed background is given according to the reviewers suggestions.

Experimental design

The revised manuscript is more explicit in describing the experimental methodology.

Validity of the findings

In general, the revised manuscript reads better with the inclusion of discussion points that add further relevance to the in silico findings. One potential controversy which may need addressing is the following;
Authors indicated that changes in metabolic pathways that are driven by oncogenes, are recognized as cancer markers; and indicated TP53 and CTNNB1 as such genes suggesting a probable route for metabolic reprogramming. Although the authors utilized a strictly metabolic gene signature, how come that signature only has predictive power in TP53 and CTNNB1 wild type patients? Would it not make more sense for such a metabolism driven signature to be a better prognostic factor in TP53 and CTNNB1 mutant HCC, where cellular metabolism is shaped by the mutant oncogene?

·

Basic reporting

Literature references, sufficient field background/context provided.

Experimental design

It is suitable.

Validity of the findings

It is suitable

Additional comments

This paper is suitable for publication.

---

## Round 0.3 · accepted · Accept

The authors are now addressed adequately all of the concerns raised by the reviewers.